# Psychopathology and Mindful Parenting in Parents of Preschool and School-Aged Children: The Role of Supportive Coparenting

**DOI:** 10.3390/ijerph20021238

**Published:** 2023-01-10

**Authors:** Daniela Mourão, Ana Fonseca, Helena Moreira

**Affiliations:** 1University of Coimbra, Faculty of Psychology and Educational Sciences, 3000-115 Coimbra, Portugal; 2University of Coimbra, Center for Research in Neuropsychology and Cognitive Behavioral Intervention, 3000-115 Coimbra, Portugal

**Keywords:** parental psychopathology symptomatology, supportive coparenting, mindful parenting

## Abstract

The present study aimed to explore the mediating role of supportive coparenting in the relationship between parental psychopathology symptoms and mindful parenting and to explore whether the child’s age group moderates the associations in the model. A sample of 462 parents (94.2% mothers) of preschool and school-aged children completed the Hospital Anxiety and Depression Scale, the Parental Perceptions of Coparenting Questionnaire, and the Interpersonal Mindfulness in Parenting Scale. A moderated mediation model was estimated to analyze the indirect effect of psychopathology symptoms on mindful parenting through supportive coparenting and the moderating role of the child’s age group. Higher levels of parental psychopathology were found to be associated both directly and indirectly (through lower levels of supportive coparenting) with lower levels of mindful parenting, regardless of the child’s age group. The results suggest that supportive coparenting is a relevant explanatory mechanism for the relationship between parental psychopathology and mindful parenting.

## 1. Introduction

Mindful parenting can be defined as a parental approach that translates into greater attention to the present moment and greater (self-)compassion in the relationship between parents and children [1,2,3]. Practicing mindful parenting enables parents to be more aware of their own internal states and of the internal states of their child during interactions and to exercise greater self-regulation in these interactions, allowing them to choose parental practices that are truly aligned with their parental values and objectives [2].

Mindful parenting can be characterized by several dimensions [2,4,5]: (1) listening with full attention, which consists of listening to the child with attention and awareness focused on the present moment that goes beyond listening only to the words that are said and allows the parent to correctly discern the behavioral cues of the child; (2) nonjudgmental acceptance of parental functioning, which involves acceptance of the traits, attributes, and behaviors of oneself and one’s child and translates into fundamental acceptance of the child as well as the establishment of clear limits and rules for the child’s behavior; (3) emotional awareness of the child, which consists of the parent’s ability to notice and identify the emotions of the child; (4) self-regulation in parenting, which involves low reactivity to the child’s behavior and less reactive/automatic discipline, which is achieved through self-control in the exercise of parental behavior; and (5) compassion for the child, which implies an attitude of kindness, sensitivity, and responsiveness to the child’s needs.

In recent years, several studies have contributed to a better understanding of the factors that impact this parental approach. For example, Moreira et al. [6] demonstrated that avoidant and anxious attachment styles can impair parents’ ability to develop a mindful parenting approach. There is some evidence that parents who have higher levels of dispositional mindfulness [7,8] and higher levels of self-compassion [6,8,9] are more able to enact mindfulness and compassion in their parenting relationship. Other important determinants of mindful parenting are emotion regulation [10], the tendency to experience negative affect [11], and parenting stress [12].

The presence of clinically significant levels of parental psychopathology symptoms seems to be a clear predictor of greater difficulties in adopting a mindful parenting approach [12,13,14]. For instance, Moreira and Canavarro [13] found that mothers of adolescents with clinically significant levels of anxiety and/or depressive symptomatology had lower levels of all mindful parenting dimensions than mothers without significant levels of anxiety and/or depressive symptomatology. In the postpartum period, higher levels of postpartum depression and anxiety were also found to be associated with lower levels of mindful parenting [12]. Although there is evidence of the negative impact of parents’ psychopathology on their ability to be mindful and compassionate in the parenting relationship, less is known about the factors that may help to explain this relationship. In the current study, we explore the mediating role of supportive coparenting.

### 1.1. Supportive Coparenting

Coparenting can be described as the way that two individuals work together to raise a child [15], assume the common task of sharing parental responsibilities and duties, and learn ways to work as a team to ensure the well-being of their children and family [16]. Research has shown that supportive coparenting is an essential factor in the quality of parenting. For example, some studies have shown that supportive coparenting is associated with higher levels of solidarity, agreement, support, and proximity between partners as well as with better communication and shared decision-making and lower levels of conflict and coparental sabotage [17,18,19]. There are also some studies, although scarce, that associate the quality of coparenting, including higher levels of supportive coparenting and a reduction in coparent conflict, with higher levels of mindful parenting [1,19].

Several studies have attempted to understand which factors can affect a supportive coparenting approach [16,20,21]. For instance, parents’ experience of anxious or depressive symptomatology seems to be a factor that can negatively affect the coparenting relationship [16]. Majdandžić et al. [20] demonstrated that parental anxiety can directly interfere with positive coparenting since anxiety can prevent a parent from fully engaging in problem solving with his or her partner. These authors suggest that it is also possible that parents’ anxiety problems result in higher levels of concern about the child’s activities, which can lead to increased coparent conflict and might affect their ability to establish clear limits and rules for the child’s behavior, one of the fundamental dimensions of a mindful parenting approach. In addition, parents’ depressive symptoms were associated with greater parental conflict and lower levels of involvement in the provision of care to their children [22], which may influence the parents’ capacity to notice the child’s emotions and, as a result, affect their ability to respond to the child’s needs, both of which are critical dimensions of a mindful parenting approach.

### 1.2. The Present Study

As mentioned above, research has shown that parental psychopathology symptoms are associated with a lower ability to exercise mindful parenting [13,23]. Although there is some evidence that parents’ mental health can also influence the quality of coparenting and that it may have an impact on the parents’ ability to adopt a mindful approach to parenting, to the best of our knowledge, no study has explored the mediating effect of coparenting on the association between parents’ mental health and mindful parenting. In addition, most studies focus on a specific age range and do not investigate whether the relationship between parents’ mental health and mindful parenting or between parents’ mental health and supportive coparenting varies according to the different developmental stages of the child.

Thus, the present study aims to explore the mediating role of supportive coparenting in the relationship between parental psychopathology symptoms and mindful parenting among parents of toddlers and preschool and grade-school children. We expect that higher levels of parental psychopathology symptoms are associated with lower levels of mindful parenting [13,23] and that this relationship is mediated by lower levels of supportive coparenting [16,20,22].

## 2. Materials and Methods

### 2.1. Participants

The sample consisted of 462 Portuguese parents (94.2% women) aged between 23 and 55 years (M = 38.92, SD = 4.90). Most parents were currently living with a partner (90.7%), were working (91.1%), had completed higher education (74.5%), reported a monthly family income up to EUR 2000 (54.5%), and lived in urban areas (81.8%), particularly in the Metropolitan Area of Lisbon (45.0%), the central region (24.0%), and the northern region (18.4%) of Portugal. The children (50.2% male) had a mean age of 6.13 years, and approximately half were school-aged children. The sociodemographic characteristics of the participants are presented in Table 1.

### 2.2. Procedures

The sample for this study was collected by convenience between May and July 2019 through an online survey hosted on the LimeSurvey^®^ website. The link that provided access to the questionnaire was shared on social networks, including a Facebook^®^ page titled “The expression of emotions in the family context”. The inclusion criteria for the study consisted of (1) being a parent of at least one child of preschool or school age and (2) being fluent in Portuguese. Parents with more than one child in the specified age range were instructed to select one child when answering the questions from the Interpersonal Mindfulness in Parenting Scale.

The first page of the online questionnaire contained information about the study’s inclusion criteria and objectives. Participants were also informed that their participation in the study was anonymous, and that no identifiable information would be collected. Participation in the study was voluntary, and participants were not given any monetary or other type of compensation. An informed consent form was presented on the second page of the online survey, and only parents who agreed to the study conditions and completed all the assessment instruments were included in this study. The present study was approved by the Ethics Committee of the Faculty of Psychology and Educational Sciences of the University of Coimbra.

### 2.3. Measures

#### 2.3.1. Parental Psychopathology Symptomatology

The Hospital Anxiety and Depression Scale [24,25] is a self-response measure that aims to assess anxious and depressive symptomatology over the past seven days. The scale includes 14 items and consists of two subscales with independent scores: (1) Anxiety (e.g., “Worrying thoughts go through my mind”) and (2) Depression (e.g., “I feel as if I am slowed down”). Items are answered on a four-point Likert scale ranging from 0 to 3. Scores for each subscale and a total score can be obtained by averaging the scores of the items; the higher the scores obtained, the higher the level of anxious and depressive symptomatology. The Portuguese version of the HADS [25] demonstrated construct validity and internal consistency, with adequate Cronbach’s alpha values in each subscale. The Cronbach’s alpha value for the total score of the scale in this study is presented in Table 2.

#### 2.3.2. Coparenting Quality

The Parental Perceptions of Coparenting Questionnaire (PCPQ-PT; Moreira and Fonseca, 2019) [26] is a self-response questionnaire that assesses parents’ perceptions of the quality of their coparenting relationship. Before completing the PCPQ-PT, parents were given the following instruction: “If your current partner is not the father/mother of your child and/or if you do not currently have a partner, please answer the questions thinking about how you interact with your child’s father/mother even if you do not live with him/her”. The questionnaire includes 14 items that assess supportive coparenting behaviors (e.g., “My partner criticizes my parenting in front of our child” and “My partner and I use similar parenting techniques”). Participants indicated the degree to which they agreed with each item while considering how often the described situation occurred. The items were answered on a five-point Likert scale ranging from 1 (never) to 5 (always), and the total score was obtained by calculating the average of the scores of all the items after reversing the scores of the items corresponding to nonsupportive coparenting behaviors. The higher the total score, the higher the level of supportive coparenting. Table 2 shows the Cronbach’s alpha value for the total score of the questionnaire in this study.

#### 2.3.3. Mindful Parenting

The instrument used to assess a mindful parenting approach was the Interpersonal Mindfulness in Parenting Scale (IM-P) [2,4]. The Portuguese version consists of 29 items (e.g., “I tend to be hard on myself when I make mistakes as a parent”; “I often react too quickly to what my child says or does”) and is organized into five factors: (1) Acceptance of Parental Functioning; (2) Self-Regulation in Parenting; (3) Compassion for the Child; (4) Listening with Full Attention; and (5) Emotional Awareness of the Child. Items are answered on a five-point Likert scale ranging from 1 (never true) to 5 (always true). After reversing some items, the total score is obtained by adding the scores of all the items, with higher scores indicating higher levels of mindful parenting. The Portuguese version [4] demonstrated construct validity and internal consistency, with adequate Cronbach’s alpha values in the respective subscales. Table 2 shows the Cronbach’s alpha value for the total scale score in this study.

### 2.4. Data Analyses

All data analyses were conducted using the Statistical Package for the Social Sciences (SPSS, version 26.0; IBM SPSS, Chicago, IL, USA) and the PROCESS computation tool [27]. Descriptive statistics were calculated for all sociodemographic variables and all study variables. Pearson’s bivariate correlations between the study variables were also calculated. Univariate analyses of variance (ANOVAs) were used to compare parents of preschool and school-aged children and parents living with and without a partner on the study variables. To identify possible covariates to be introduced in the moderated mediation model, Pearson’s bivariate correlations between some sociodemographic variables and the mediator and dependent variables were also calculated. Cohen’s guidelines [28] were used to describe and interpret the effect sizes of correlations (i.e., weak for correlations close to 0.10, moderate for correlations close to 0.30, and strong for correlations of 0.50 or higher).

First, we estimated a moderated mediation model (Model 59) [27] to explore the possible moderating role of children’s age group (preschool and school-aged children) in the relationships between parental psychopathology symptoms (independent variable; IV), supportive coparenting (mediator; M), and mindful parenting (dependent variable; DV). Given that no significant interaction was found, a simple mediation model was estimated (Model 4) [27], controlling for some covariates that were significantly correlated with the mediator or dependent variables. Indirect effects were calculated using a bootstrapping procedure (10,000 samples), which creates bootstrap 95% confidence intervals of the indirect effects. An indirect effect was considered significant if zero was not contained within the lower and upper limits of the CIs.

## 3. Results

### 3.1. Preliminary Analyses

The descriptive statistics and Pearson’s bivariate correlations for the variables under study are presented in Table 2. Parental psychopathology symptomatology was negatively and significantly correlated with supportive coparenting and with mindful parenting, and supportive coparenting was positively and significantly correlated with mindful parenting. All correlations were moderate, except for the strong correlation between parental psychopathology symptomatology and mindful parenting. The Pearson’s bivariate correlations between supportive coparenting, mindful parenting, and some sociodemographic variables were also analyzed to determine whether any variable should be introduced in the moderated mediation model as a covariate. As is shown in Table 3, negative and significant correlations were found between supportive coparenting and parents’ current marital status and between mindful parenting and the number of children. Additionally, a positive and significant correlation was found between supportive coparenting and monthly household income.

No significant differences were found between parents of preschool children and school-aged children in psychopathology (F(1, 460) = 0.13, *p* = 0.72), mindful parenting (F(1, 460) = 0.12, *p* = 0.73), or supportive coparenting (F(1, 460) = 1.98, *p* = 0.16). Likewise, no significant differences were found in psychopathology (F(1, 460) = 0.92, *p* = 0.34) or mindful parenting (F(1, 460) = 0.85, *p* = 0.36) between parents living with or without a partner. A significant difference was found in supportive coparenting (F(1, 460) = 58.01, *p* < 0.001), with parents living with a partner (M = 4.17, SD = 0.56) presenting higher levels of supportive coparenting that those living without a partner (M = 3.44, SD = 0.92).

### 3.2. Moderated Mediation and Simple Mediation Analysis

The first model explored was a moderated mediation model in which the child’s age group was the moderator. Current marital status, the number of children, and monthly household income were introduced as covariates in the model. No significant interaction was found between the moderator and any of the variables (anxious/depressive symptoms × age group in the path linking the IV to the M: b = 0.00, *p* = 0.81; anxious/depressive symptoms × age group in the path linking the IV to the DV: b = −0.01, *p* = 0.92; and supportive coparenting × age group in the path linking the M to the DV: b = −0.45, *p* = 0.76).

Because no significant interaction was found, the moderator was excluded from the model, and a simple mediation model that controlled for the same covariates was tested. As is shown in Figure 1, there was a negative and significant association between parental psychopathology symptomatology and supportive coparenting (b = −0.24, SE = 0.00, *p* < 0.001, 95% CI = [−0.03, −0.01]) in a model that explained 17.68% of supportive coparenting variance (F(4, 457) = 24.53, *p* < 0.001). In turn, supportive coparenting was positively and significantly associated with mindful parenting (b = 0.21, SE = 0.76, *p* < 0.001, 95% CI = [2.19, 5.20]) in a model that explained 28.88% of the mindful parenting variance (F(5,456) = 37.03, *p* < 0.001). As is shown in Figure 1, the direct effect of parental psychopathology symptomatology on mindful parenting was found to be significant (b = −0.43, SE = 0.07, *p* < 0.001). Additionally, a significant indirect effect of parental psychopathology symptomatology on mindful parenting was found through supportive coparenting (indirect effect = −0.05, SE = 0.02, 95% CI = [−0.08, −0.02]).

## 4. Discussion

The present study sought to examine the mediating role of supportive coparenting in the relationship between parental psychopathology symptoms and mindful parenting among parents of preschool children and school-aged children. Overall, the results corroborate the initially established hypotheses and show that the associations explored in the present study do not vary among parents of children in distinct developmental stages.

First, as expected, a significant direct effect of parental psychopathology symptoms on mindful parenting was found; that is, parents with higher levels of psychopathology symptoms seem to have increased difficulty exerting mindful parenting. These results are congruent with previous studies that have found a significant negative association between parental psychopathology and mindful parenting [13,23]. These results can be explained by analyzing how parental psychopathology relates to each of the dimensions of mindful parenting.

First, parents’ ability to listen to their children with focused attention and awareness may be diminished when anxious and/or depressive parental symptoms are present since these symptoms can lead to a greater focus on the parents’ own needs [29,30], which may result in increased difficulty directing their complete attention to their children [23]. This greater focus on themselves can also decrease the parents’ ability to be compassionate toward their children and impair their ability to notice and identify their children’s emotions, which would allow them to make conscious (versus reactive) choices about how to behave in parent–child interactions.

Second, nonjudgmental acceptance of the traits, attributes, and behaviors of the child and of themselves as parents may be diminished in anxious and/or depressed parents as a result of the increased tendency of these parents to self-criticize [31,32]. It is well known that self-criticism is a specific cognitive process of depression [33,34], and therefore it is not surprising that depressed parents may also criticize themselves in their parenting roles.

Third, parental self-regulation may be compromised in parents who experience psychopathological symptoms given the association between emotional dysregulation and psychopathology [35,36]. Thus, parents with higher levels of anxious/depressive symptomatology may have greater difficulty regulating their emotions and behaviors in the context of parenting, which can lead to greater reactivity when they are confronted with their child’s negative behaviors [13]. This may prevent them from adopting a parenting approach that is congruent with their values and objectives as parents.

As expected, we also found a significant indirect effect of parental psychopathology symptoms on mindful parenting through supportive coparenting. These results suggest that parents with higher levels of symptomatology tend to perceive less supportive coparenting, which in turn seems to compromise their ability to adopt a mindful parenting approach. The negative association between parental psychopathology symptoms and supportive coparenting is consistent with previous studies [16,20,22] and may be explained by several factors. First, parental psychopathology may affect the ability of one parent to exhibit supportive behaviors toward the partner without psychopathology. As an example, the presence of anxiety in a parent may be associated with concern that the other partner encourages the child’s autonomy due to the perceived risks that this autonomy can represent. Thus, high levels of anxiety in one parent may lead to sabotage of the nonanxious partner’s parenting behaviors rather than attempts to balance different parenting approaches [20]. Another aspect that can enhance sabotage behaviors by the partner with psychopathology, such as criticizing and blaming the other parent [37], may be the perception of low self-efficacy and low parental efficacy. By deprecating or criticizing the partner’s parental behaviors, parents with psychopathology share their perception of low parental efficacy with the other parent, which can diminish their perception of guilt that may be associated with their experience of anxious and depressive symptoms.

Second, the presence of anxious and/or depressive symptoms may lead to a lower tolerance of differences in parental approaches, which may increase parental disagreements about child-related issues. Additionally, a parent with anxious and/or depressive symptomatology may tend to avoid anxiety-provoking stimuli or distress (e.g., conflicts related to disagreement over parental strategies) and, consequently, withdraw from coparenting interactions [20]. Third, psychological inflexibility, which is often associated with the presence of psychopathology [35,38,39,40], may be an important factor in the decrease in supportive coparenting behaviors given the association between how parents manage the division of child-related tasks and the degree of flexibility versus rigidity that parents present in this division [37]. For instance, parents with psychopathology may have very strict rules about who should do what, while parents without clinically significant levels of anxious and/or depressive symptoms may approach tasks more flexibly, adjusting responsibilities as situations arise.

The results also demonstrate that higher levels of supportive coparenting are associated with higher levels of mindful parenting. Research on this association is still scarce, but some preliminary results suggest that a mindful approach to parenthood can improve the quality of the coparenting relationship [1,19]. The results obtained in this study suggest that supportive coparenting can also promote a mindful parenting approach. In fact, higher levels of solidarity, support, and proximity between partners [17,18,19] may contribute to a greater capacity to endure the intense emotions that frequently arise in the exercise of parenting [2] and to greater self-control of parental behaviors since parents feel that they can share these experiences with another who will understand the difficulties and provide support. A supportive coparenting relationship can also contribute, through empathy and support, to increased compassion among partners, which may facilitate the acceptance of one’s attributes and behaviors as a parent and promote greater self-compassion when parental goals are not achieved.

In addition, the fact that supportive coparenting is associated with better communication between partners, shared decision-making, and lower levels of conflict [17,18,19], including in the presence of the child, may facilitate the joint establishment of clear rules and limits for children’s behavior, which is also an important factor in mindful parenting. Furthermore, in a supportive coparenting relationship, child-related decisions are shared between partners, and this may contribute to lower levels of individual guilt when parental objectives are not met, thereby increasing the nonjudgmental acceptance of the self as a parent. Finally, higher agreement between partners and lower levels of coparenting sabotage [17,18,19] may enable parents to feel more secure in their parenting behavior and the discipline applied to their children, and to self-criticize and blame themselves less frequently in their relationship with their children.

### Limitations and Future Research

This study includes some limitations that should be mentioned. First, as a cross-sectional study, it does not allow the direction between the variables to be established. Although the relationship suggested in our mediation model is supported by the literature, the relationship can be bidirectional. It will be important to implement longitudinal or experimental studies in the future, as these would make it possible to determine the direction between the variables with more certainty. Second, the representativeness of the sample may be compromised due to the smaller number of male participants (5.8%) compared with the number of female participants (94.2%). In this sense, it will be relevant for future investigations to analyze the relationships studied in a sample consisting of an equivalent number of mothers and fathers. Third, the sample was collected exclusively online, which may have led to self-selection bias and may have compromised the representativeness of the sample. Finally, the exclusive use of self-response instruments may have compromised the validity of the results through the possible influence of social desirability factors. Additionally, the self-response instruments used do not allow the clinical diagnosis of anxious and/or depressive disorders. Thus, it is important for future studies to use semistructured clinical interviews for greater reliability and validity of the results.

## 5. Conclusions

Despite its limitations, this study represents an innovative contribution to research and clinical practice in the context of parenting. The results of this study allow us to understand the important role of supportive coparenting in the way parents with high levels of symptoms of anxiety or depression exercise their parenting. By identifying a modifiable mediating factor (supportive parenting), this study suggests that an intervention aimed at promoting mindful parenting for parents of preschool and school-aged children who experience anxiety and/or depression problems should be implemented in the family subsystem to promote supportive behaviors between partners. Partners who disagree on essential aspects related to the child’s upbringing, who do not support each other, or who sabotage the other will rarely be able to integrate into their parenting the important characteristics of mindful parenting, such as self-regulation or compassion for the child, particularly if one of the parents experiences psychopathological symptomatology.

## Figures and Tables

**Figure 1 ijerph-20-01238-f001:**
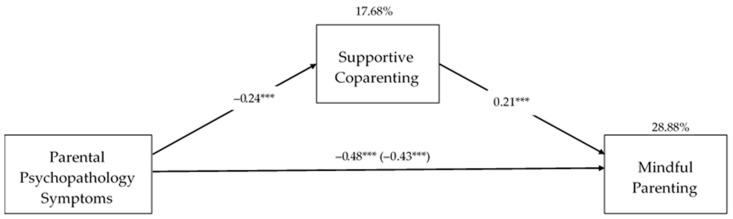
Statistical diagram of the simple mediation model between parental psychopathology symptoms and mindful parenting through supportive coparenting. *** *p* < 0.001. *Note*: The values of the trajectories represent the standardized regression coefficients. In the arrow that links parental psychopathology symptomatology to mindful parenting, the value outside the parentheses represents the total effect of parental psychopathology symptomatology on mindful parenting. The value within the parentheses represents the direct effect of parental psychopathology symptomatology on mindful parenting after the inclusion of supportive coparenting.

**Table 1 ijerph-20-01238-t001:** Sociodemographic Characteristics of Participants.

Parents	*N* = 462
**Gender**	
Female (*n* (%))	435 (94.2%)
Male (*n* (%))	27 (5.8%)
Age (*M* (*SD*); range)	38.92 (4.90); 23–55
**Marital status**	
Lives with a partner (*n* (%))	419 (90.7%)
Lives without a partner (*n* (%))	43 (9.3%)
**Professional situation**	
Working (*n* (%))	421 (91.1%)
Unemployed or student (*n* (%))	41 (8.9%)
**Education**	
Primary or secondary education (*n* (%))	118 (25.5%)
Higher education (*n* (%))	344 (74.5%)
**Monthly household income**	
Up to EUR 2000 (*n* (%))	252 (54.5%)
More than EUR 2001 (*n* (%))	210 (45.5%)
**Residence**	
Urban area (*n* (%))	378 (81.8%)
Rural area (*n* (%))	84 (18.2%)
**NUTS2**	
Lisbon Metropolitan Area (*n* (%))	208 (45.0%)
Central region (*n* (%))	111 (24.0%)
Northern region (*n* (%))	85 (18.4%)
Alentejo (*n* (%))	17 (3.7%)
Algarve (*n* (%))	10 (2.2%)
Autonomous Region of the Azores (*n* (%))	7 (1.5%)
Autonomous Region of Madeira (*n* (%))	5 (1.1%)
Foreigner (*n* (%))	19 (4.1%)
**Children**	
**Gender**	
Female (*n* (%))	230 (49.8%)
Male (*n* (%))	232 (50.2%)
Age (*M* (*SD*); range)	6.13 (3.20); 1–13
**Age group**	
Preschool children (1–5 years) (*n* (%))	221 (47.8%)
School-aged children (5–13 years) (*n* (%))	241 (52.2%)

*Note.* Regions of Portugal were defined according to the Portuguese Nomenclature of Territorial Units for Statistics 2 (NUTS-2).

**Table 2 ijerph-20-01238-t002:** Descriptive Statistics and Correlations between the Variables Under Study.

Variables under Study	*Range* (*DP*); Min–Max	α	1	2
1. Psychopathology Symptomatology	11.85 (6.60); 0–42	0.89	-	
2. Supportive Coparenting	57.46 (8.95); 14–70	0.92	−0.25 **	-
3. Mindful Parenting	104.94 (11.22); 29–145	0.89	−0.49 **	0.30 **

** *p* < 0.01.

**Table 3 ijerph-20-01238-t003:** Correlations between sociodemographic variables and the mediator and dependent variables.

	Supportive Coparenting	Mindful Parenting
Age	–0.03	0.01
Gender	–0.04	–0.03
Current marital status	–0.34 **	–0.04
Education	–0.08	0.01
Current professional situation	–0.01	–0.02
Monthly household income	0.18 **	0.01
Number of children	0.04	–0.14 **
Child’s age	–0.07	–0.01
Child’s gender	–0.06	–0.08

** *p* < 0.01. *Note*. Gender: 1 = Female, 2 = Male; Current marital status (“Do you currently live with a partner?”): 1 = Yes, 2 = No; Education: 0 = High school or below, 1 = Higher education; Current professional situation: 1 = Working, 2 = Not working; Monthly household income: 0 = Up to EUR 2000, 1 = EUR 2001 or above; Child’s gender: 1 = Female, 2 = Male.

## Data Availability

The data presented in this study are available on request from the corresponding author. The data are not publicly available due to ethical reasons.

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
