# Peer review of "Psychopathology and Mindful Parenting in Parents of Preschool and School-Aged Children: The Role of Supportive Coparenting"

_ijerph, 2023, doi:10.3390/ijerph20021238_

Round 1

Reviewer 1 Report

The piece describes a curious study in which the aim was to explore the mediating role of supportive coparenting in the relationship between parental psychopathology symptoms and mindful parenting and to explore whether the child’s age group moderates the associations in the model. The abstract is an objective representation of the article. The review of the literature mentions a number of relevant studies some of them are very recent and the others are classic works. Also, the introduction highlights why it is important the study. The materials and methods are described with sufficient detail. The results section provides a precise description of the experimental results and authors discuss the results clearly. The future research directions are mentioned. However, there are a number of problems that recommend this article be revised before it should be considered for publication:

- Participants: How were the participants selected? It is necessary to indicate that they were selected for convenience.

- Some numbers are expressed with two decimal places and others with three. It would be convenient to unify criteria. The comma is should be used to separate decimals; although, in this article is used the point.

- References: The rules of the International Journal of Environmental Research and Public Health has not been respected.

I hope that these comments, oriented toward formative feedback, will help the authors to improve the text. Good job.

Author Response

The piece describes a curious study in which the aim was to explore the mediating role of supportive coparenting in the relationship between parental psychopathology symptoms and mindful parenting and to explore whether the child’s age group moderates the associations in the model. The abstract is an objective representation of the article. The review of the literature mentions a number of relevant studies some of them are very recent and the others are classic works. Also, the introduction highlights why it is important the study. The materials and methods are described with sufficient detail. The results section provides a precise description of the experimental results and authors discuss the results clearly. The future research directions are mentioned. However, there are a number of problems that recommend this article be revised before it should be considered for publication:

- Participants: How were the participants selected? It is necessary to indicate that they were selected for convenience.

REPLY: this information was added in the Procedures section

- Some numbers are expressed with two decimal places and others with three. It would be convenient to unify criteria. The comma is should be used to separate decimals; although, in this article is used the point.

REPLY: the paper follows the APA publication norms (7th edition). Although three decimal places can be used to report p values, we followed the reviewer’s suggestion and presented it with two decimal places. According to the APA manual, a period, not a comma, should separate decimal numbers. Therefore, we decided to maintain the results presentation as it is.

- References: The rules of the International Journal of Environmental Research and Public Health has not been respected.

REPLY: the References section has been amended and is now in compliance with the International Journal of Environmental Research and Public Health's guidelines.

I hope that these comments, oriented toward formative feedback, will help the authors to improve the text. Good job.

REPLY: thank you very much for the encouragement!

Reviewer 2 Report

The study is really good. The only concern is actualize the references because is a study had made in 2019 and scientific literature is growing each day. It will be desirable more recent references. 

Another point that would have to be reviewed in the conclusions, including future perspectives and the limitations of the study that were further developed and how they can be addressed.

Author Response

The study is really good. The only concern is actualize the references because is a study had made in 2019 and scientific literature is growing each day. It will be desirable more recent references. 

REPLY: Following the reviewer's advice, we updated the content with a few more recent references. This changed the numerical order of all the remaining references, which were updated in the References section.

Another point that would have to be reviewed in the conclusions, including future perspectives and the limitations of the study that were further developed and how they can be addressed.

REPLY: We list a number of the study's limitations and offer recommendations for how to overcome them near the end of the Discussion, immediately before the Conclusions section. We are unsure if the Reviewer is interested in additional information.

Reviewer 3 Report

This article presents findings from a longitudinal analysis of national survey data examining the relationship between self-reported exposure to types of child maltreatment and self-reported substance use in emerging adulthood. It is a well-organized and well-written manuscript using an innovative regression-based analytic technique to understand risk trajectories over time. Although it has the potential to make a nice contribution to the literature, several critical methodological issues are unaddressed. The discussion is also underdeveloped.

1. In the introduction, the authors should consider further illustration of how supportive coparenting plays a moderating role in the association between parental psychopathology and mindful parents. Although the authors briefly mentioned it in the last paragraph before the present study, it is helpful for readers to understand better that the potential mechanisms connect parental psychopathology to mindful parenting through coparenting.

2. The main variable, coparenting, requires the participating parents to have partners so that coparenting can be measured (e.g., in your measure: My partner criticizes my parenting in front of our child). However, there are 43 parents without partners. Please indicate whether these parents are included in the main analyses and if so, the authors can consider removing them.

3. Since the authors mentioned that no compensation was paid for participation, I wonder if this would influence the sample characteristics or willingness to participate.

4. Figure 1 has several errors in the graph (e.g., what is 258, 257, and the 2… above coparenting).

5. Could the authors provide standardized coefficients in the mediation analyses? These can provide more straightforward interpretations. 

Author Response

Response to Reviewer #3

  1. In the introduction, the authors should consider further illustration of how supportive coparenting plays a moderating role in the association between parental psychopathology and mindful parents. Although the authors briefly mentioned it in the last paragraph before the present study, it is helpful for readers to understand better that the potential mechanisms connect parental psychopathology to mindful parenting through coparenting.

REPLY: Following the reviewer's suggestion, the last paragraph before the current study section now includes a better explanation of some of the potential mechanisms linking parental psychopathological symptoms to a mindful parenting approach through supportive coparenting.

  1. The main variable, coparenting, requires the participating parents to have partners so that coparenting can be measured (e.g., in your measure: My partner criticizes my parenting in front of our child). However, there are 43 parents without partners. Please indicate whether these parents are included in the main analyses and if so, the authors can consider removing them.

REPLY: This is a very important question. In fact, 43 participants were not living with a partner at the time they participated in the study. However, before completing the PCPQ-PT, parents had to read the following instruction: “If your current partner is not the father/mother of your child and/or if you do not currently have a partner, please answer the questions thinking about how you interact with your child's father/mother even if you do not live with him/her”. This issue is now clear in the description of the PCPQ-PT.

  1. Since the authors mentioned that no compensation was paid for participation, I wonder if this would influence the sample characteristics or willingness to participate.

REPLY: this is a very interesting consideration. In Portugal, nevertheless, it's uncommon to compensate people for their participation, especially in online surveys. As a result, we think that this variable has no discernible influence on the study's findings.

  1. Figure 1 has several errors in the graph (e.g., what is 258, 257, and the 2… above coparenting).

REPLY: These numbers appear to be line numbers that are misplaced; they were not added by the authors and therefore should be omitted in the published version of the manuscript.

  1. Could the authors provide standardized coefficients in the mediation analyses? These can provide more straightforward interpretations. 

REPLY: Standardized numbers are now presented in the results and in the figure depicting the model.